# Single Center Experience of Oral Vancomycin Therapy in Young Patients with Primary Sclerosing Cholangitis: A Case Series

Amala J. Alenchery [1,*,†], Sophia Patel [1], Lori Mahajan [1], Jacob A. Kurowski [1], Sarah Worley [2], Vera Hupertz [1], Kaddakal Radhakrishnan [1] and Mohammad Nasser Kabbany [1,†]

[1] Department of Pediatric Gastroenterology, Hepatology and Nutrition, Cleveland Clinic, 9500 Euclid Avenue, Cleveland, OH 44195, USA
[2] Pediatric Research Institute, Cleveland Clinic, 9500 Euclid Avenue, Cleveland, OH 44195, USA
* Correspondence: alencha@ccf.org; Tel.: +1-216-445-5555; Fax: +1-216-444-2974
† These authors contributed equally to this work.

**Abstract:** There is no single proven therapy that prolongs hepatic transplant-free survival in patients with primary sclerosing cholangitis (PSC). Oral vancomycin (OV) has shown some benefit in small pediatric and adult series. We describe the effect of OV on pediatric onset PSC at our tertiary hospital. This is a single-center, retrospective, descriptive case series involving patients (<21 years at diagnosis) with PSC on OV from 2001 till 2021. The therapy effect was assessed based on symptoms, biochemical labs, imaging and liver biopsy at six and twelve months, and then annually until therapy was discontinued. The inclusion criteria identified 17 patients. Baseline GGT (n = 17) was elevated among 88.2% which then normalized among 53.8% (n = 13) at six months and 55.6% (n = 9) at one year post-OV. Baseline ALT normalized in 58.8% (n = 17) at six months and 42.8% (n = 14) at one year. Imaging findings within one year of OV revealed improved/stable biliary findings among 66.7% (n = 8/12). No adverse events were reported. OV was associated with an improvement in bile duct injury marker (GGT) after at least six months of therapy, with no disease progression on imaging within one year of therapy.

**Keywords:** primary sclerosing cholangitis; oral vancomycin; pediatric liver disease; therapy

## 1. Introduction

### 1.1. Background

Primary sclerosing cholangitis (PSC) is an autoimmune progressive disorder characterized by chronic diffuse inflammation as well as stricturing and fibrosis of extra- and/or intrahepatic bile ducts. This disorder ultimately progresses to liver cirrhosis and end-stage liver disease (ESLD) in a vast majority of the affected population [1,2]. There is a complex inter-play between environmental and genetic factors in the disease process. The exact mechanism behind the pathogenesis of PSC is unclear; however, there is evidence of marked dysbiosis and decreased diversity of the gut microbiome among these patients compared to healthy controls and patients with inflammatory bowel disease (IBD) [1,3]. The course of PSC is often complicated by portal hypertension, infectious cholangitis, and hepatobiliary malignancy, among others, leading to decreased quality of life and survival with need for increased healthcare utilization [4,5].

Improved screening along with widespread knowledge about the disease process has led to increasing diagnosis of PSC among children and adolescents with concomitant IBD, affecting at least 10% of children with ulcerative colitis (UC). Conversely, at least 70% patients with PSC already have or will develop IBD [6]. The exact relationship between IBD and PSC is not well understood but is postulated to be related to the altered gut flora resulting in immune mediated destruction of bile ducts [5]. Pediatric PSC commonly presents with one or more symptoms including abdominal pain, diarrhea, fatigue, anorexia,

and pruritus [7]. Compared to elevated alkaline phosphatase (ALP) levels in adults [8], elevated gamma glutamyl transferase (GGT) levels among children with PSC are considered important biomarkers to assess disease status [7,9,10]. There is great variability in ALP among children, with significant increase during periods of rapid bone growth, making it less reliable as a marker for pediatric liver disease [7].

Currently, there is no established medical therapy to manage the progression and recurrence of this chronic, debilitating disorder. Approximately 30% children with PSC ultimately require liver transplantation [1,5,11]. There are several emerging therapies for PSC including bile-acid based therapy such as ursodiol, immunosuppressants/ immunomodulators and microbiota-based therapy such as antibiotics and fecal transplantation in the adult population [12,13]. Several adult and pediatric studies have evaluated cholagogues such as ursodiol and antibiotics such as oral vancomycin (OV) with some identified benefits [11,14]. Oral vancomycin, a tricyclic glycopeptide antibiotic, with bactericidal activity against gram-positive bacteria, is FDA-approved for *Clostridium difficile*-associated diarrhea [15]. Off-label use of OV has been described in literature as a promising option for management of PSC, however there is lack of evidence on its effect on slowing or halting the disease course. The postulated mechanism of action is by altering gut microbiota and having local anti-inflammatory properties secondary to its immunomodulatory effects and possible effects on regulatory T cells, compounded by its favorable side effect profile, highlighting its potential as a safe long-term therapeutic option [12,16]. In this case series, we describe the experience with OV therapy in young patients with PSC at our tertiary level hospital.

### 1.2. Objectives

This is a retrospective, single center study undertaken with the objective to describe the effect of OV therapy on PSC in children and young adults at our tertiary care center.

## 2. Materials and Methods

### 2.1. Sample and Setting

This study included all patients who were younger than 21 years of age when diagnosed with primary sclerosing cholangitis, and who received oral vancomycin therapy from the inception of electronic medical records at our tertiary-level pediatric hospital, i.e., August 2001 to April 2021. Diagnosis of PSC was made with findings of characteristic bile duct changes on imaging by either magnetic resonance cholangiography (MRCP) and/or endoscopic retrograde cholangiopancreatography (ERCP) in all patients with or without liver biopsy. We excluded patients on oral vancomycin therapy for other conditions such as pouchitis and recurrent *Clostridium difficile* (C. diff) infection.

### 2.2. Baseline Data

The patient characteristics described include demographics such as gender and ethnicity, age at diagnosis, concomitant diseases (IBD and autoimmune hepatitis), time from PSC diagnosis to initiation of OV (see Table 1), duration and dose of OV therapy, concomitant pharmacologic therapies as well as identified adverse effects.

Clinical, biochemical, imaging and histopathology data was collected at baseline, prior to initiation of OV, at six months and 12 months after initiation of therapy. The data parameters were then followed every 12 months until "the most recent follow up" to identify the disease course as well as medication trials for PSC. Laboratory parameters obtained include gamma glutamyl transferase (GGT), alkaline phosphatase (ALP), alanine transaminase (ALT), aspartate aminotransferase (AST), albumin, total bilirubin (TB), complete blood count (CBC) including white blood cell (WBC) count, hemoglobin (Hb), hematocrit (HCT), platelet count (PLT), C-reactive protein (CRP) and erythrocyte sedimentation rate (ESR) when available. Liver ultrasound (US) and magnetic resonance cholangiopancreatography/imaging (MRCP/MRI) of the abdomen were utilized as imaging modalities. Liver biopsy and histopathology findings were also obtained, when available.

**Table 1.** Baseline characteristics of young patients with primary sclerosing cholangitis prior to initiation of oral vancomycin therapy.

| Patient Number | Age at Diagnosis (Year) | Gender | Associated Diagnosis- CD/UC | Time from Diagnosis to OV | Albumin (g/dL) | Total Bilirubin (mg/dL) | GGT (U/L) | ALT (U/L) | AST (U/L) | ALP (U/L) | ESR (mm/Hour) |
|---|---|---|---|---|---|---|---|---|---|---|---|
| Patient 1 | 5 | Male | UC | 4 years 8 months | 3.6 | 1.2 | **122** | **85** | **72** | **467** | 17 |
| Patient 2 | 9 | Male | - | 5 months | 4.3 | 0.6 | **171** | **160** | **56** | **422** | - |
| Patient 3 | 11 | Male | CD | 2 years 4 months | 3.3 | 1.0 | **251** | **67** | **94** | **980** | **109** |
| Patient 4 | 13 | Male | - | 1 year 9 months | 4.1 | **1.6** | 38 | 33 | 35 | **444** | 24 |
| Patient 5 | 13 | Male | CD | 3 years 9 months | 4.2 | 0.5 | **512** | **146** | **96** | **500** | 19 |
| Patient 6 | 16 | Female | UC | 1 month | 4.3 | 0.2 | **155** | 38 | 44 | 200 | **37** |
| Patient 7 | 16 | Male | CD | 10 days | 4.7 | 0.8 | **720** | **275** | **140** | 274 | 9 |
| Patient 8 | 16 | Male | CD | 13 days | 2.8 | **1.7** | **198** | **106** | **100** | **661** | 87 |
| Patient 9 | 17 | Male | UC | 6 days | 4.4 | 0.3 | **120** | 30 | 17 | 128 | **35** |
| Patient 10 | 18 | Male | UC | 7 days | 4.1 | 0.6 | **381** | **252** | **124** | **383** | 10 |
| Patient 11 | 13 | Male | UC | 5 years | 3.8 | 0.5 | **275** | **84** | **65** | **677** | 19 |
| Patient 12 | 18 | Male | UC | 8 months | 4.2 | 0.2 | 21 | 12 | 22 | 160 | 14 |
| Patient 13 | 18 | Male | UC | 1 year 10 months | 4.4 | 0.6 | **215** | 31 | 33 | 143 | 31 |
| Patient 14 | 20 | Male | UC | 4 years 7 months | 3.9 | **1.8** | **443** | **72** | **108** | 294 | 24 |
| Patient 15 | 22 | Female | UC | 9 years | 4.0 | 0.4 | **221** | **53** | **45** | 248 | 25 |
| Patient 16 | 22 | Female | UC | 7 months | 2.4 | **1.5** | **627** | **31** | **93** | **845** | **100** |
| Patient 17 | 16 | Male | UC | 2 years 10 months | 4.4 | 0.5 | **139** | **97** | **99** | 124 | 1 |

CD—Crohn's disease, UC—Ulcerative colitis, OV—oral vancomycin, GGT—gamma glutamyl transferase, ALT—alanine aminotransferase, AST—aspartate aminotransferase, ALP—alkaline phosphatase, ESR—erythrocyte sedimentation rate.

*2.3. Data Collection and Analysis*

The study data were collected, de-identified, reviewed, and managed using REDCap (Research Electronic Data Capture) electronic data capture tools hosted at our institution in compliance with the Health Insurance Portability and Accountability Act (HIPAA) regulations. The Institutional Review Board (IRB) at our hospital deemed the study to be exempt research, i.e., minimal risk research using/involving secondary research which does not require consent and waived the need for continuing review.

The data is described using medians, ranges, or means and standard deviations for continuous variables and percentages for categorical variables. Changes in pertinent laboratory values from baseline to six months and one year following initiation of OV were assessed using two-sided Wilcoxon signed rank tests with a significance criterion of 0.05.

**3. Results**

*3.1. Demographics*

A total of 17 patients were retrospectively identified per the inclusion and exclusion criteria of our study (Table 1). The diagnosis of PSC was made based on characteristic MRCP findings in all patients, with liver histopathology reports available for seven patients. We identified 14 males and three females in the cohort resulting in a 4.7:1 proportion between males and females. The mean age at diagnosis with PSC was 15.5 years with a median of 16 years (range 5–22 years). 88.2% identified to be of Caucasian ethnicity while the remaining 11.8% patients identified as multiracial or multicultural (Table 1).

Within this 17-patient cohort, 15 patients had the concomitant diagnosis of IBD with UC identified in 73.3% (n = 11) and 26.7% with CD (n = 4). Patients 5, 8 and 11 were diagnosed with autoimmune hepatitis or PSC overlap disease. Patients 5 and 8 also had

Crohn's Disease while Patient 11 had the concomitant diagnosis of Ulcerative Colitis. There was a strong family history of PSC in Patient 4.

*3.2. Baseline Characteristics*

3.2.1. Baseline Clinical Features

We identified the predominantly associated clinical symptoms of PSC in all 17 patients during the pre-therapy stage. The baseline percentage distribution of symptoms in our cohort were fatigue (41%), pruritus (35%), jaundice (11.8%) and anorexia (41.2%) (see Figure 1).

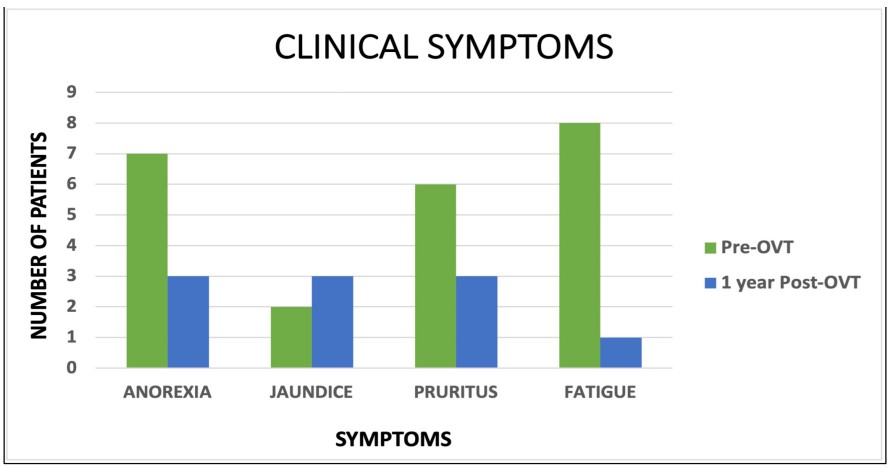

**Figure 1.** Clinical symptoms before oral vancomycin therapy (OVT) and one-year post-OVT.

3.2.2. Baseline Biochemical Laboratory Values

The mean for baseline TB was 0.82 mg/dL with values ranging from 0.2–1.8 mg/dL, median of 0.6 mg/dL (reference range for normal is 0.2–1.3 mg/dL). Baseline ALT ranged from 12–275 U/L (reference range for normal is 10–54 U/L) with a mean of 92.4 U/L and median of 72 U/L; AST range was 17–140 U/L (reference range for normal is 14–40 U/L) with a mean of 73.1 U/L and median of 72 U/L. Baseline ALP ranged from 124–980 U/L (reference range for normal is 116–468 U/L) with a median of 383 U/L and GGT ranged from 21–720 U/L (reference range for normal was 10–70 U/L) with a median of 215 U/L (see Tables 2 and 3).

3.2.3. Baseline Imaging and Histopathology Findings

Liver ultrasound (US) and magnetic resonance cholangiopancreatography/imaging (MRCP / MRI) of the abdomen were utilized as imaging modalities. Baseline liver ultrasound was not available for four patients (8, 9, 10 and 13). Among the 13 patients with US imaging, evidence of large-duct PSC was identified among 46.2% (n = 6). Baseline MRCP findings diagnostic for PSC were available for all patients. Histopathological data from liver biopsy was available for eight patients prior to initiation of OV with repeat histopathology available only in patients 4 and 9 within one year of OV therapy.

*3.3. Oral Vancomycin Regimen*

OV was initiated at a range of 0–9.1 years from PSC diagnosis with a median of 1.6 years. OV was initiated immediately following the diagnosis of PSC in 29% (n = 5) with 47% patients (n = 8) being treated within one year of diagnosis. Duration of OV was a mean of 3.04 ± 2.41 years (median of 2 years, range 0.5–8 years). The longest duration of OV was eight years (Patient 5) followed by 7.5 years (Patient 11) and seven years (Patient 1). The median age of initiation of OV therapy was 16 years. There were no adverse effects noted with OV in any of the patients including patient 5, who was on OV for a total duration

of eight years. Currently, 41.2% patients (n = 7/17) in this cohort continue to remain on OV therapy.

**Table 2.** Lab parameters (ALT, GGT) pre- and post-oral vancomycin therapy.

| Patients | GGT (U/L) | GGT 6 Months | GGT 1 Year | ALT (U/L) | ALT 6 Months | ALT 1 Year |
|---|---|---|---|---|---|---|
| Patient 1 | 122 | 60 | 17 | 85 | 65 | 55 |
| Patient 2 | 171 | 21 | - | 160 | 30 | 33 |
| Patient 3 | 251 | 254 | 312 | 67 | 56 | 62 |
| Patient 4 | 38 | 41 | 32 | 33 | 35 | 34 |
| Patient 5 | 512 | 73 | 57 | 146 | 34 | 30 |
| Patient 6 | 155 | 25 | 34 | 38 | 11 | 16 |
| Patient 7 | 720 | 33 | - | 275 | 32 | - |
| Patient 8 | 198 | - | - | 106 | 89 | 85 |
| Patient 9 | 120 | 85 | 305 | 30 | 40 | 125 |
| Patient 10 | 381 | 283 | 118 | 252 | 176 | 82 |
| Patient 11 | 275 | 328 | 204 | 84 | 327 | 167 |
| Patient 12 | 21 | 10 | 40 | 12 | 29 | 30 |
| Patient 13 | 215 | 37 | - | 31 | 12 | 8 |
| Patient 14 | 443 | 122 | - | 72 | 55 | 66 |
| Patient 15 | 221 | - | - | 53 | 32 | 70 |
| Patient 16 | 627 | - | - | 31 | 70 | - |
| Patient 17 | 139 | - | - | 97 | 18 | - |
| Median | **215** | **60** | **57** | **72** | **35** | **58.5** |
| N= | 17 | 13 | 9 | 17 | 17 | 14 |
| Range | 21–720 | 10–328 | 17–312 | 12–275 | 11–327 | 8–167 |

ALT—alanine aminotransferase and GGT—gamma glutamyl transferase labs at baseline, six months and one year after OV therapy.

**Table 3.** Lab parameters for all included patients at baseline, six months and one year of oral vancomycin therapy.

| | Baseline | | 6 Months | | 1 Year | |
|---|---|---|---|---|---|---|
| **Factor** | **N** | **Median (min, max)** | **N** | **Median (min, max)** | **N** | **Median (min, max)** |
| Total Bilirubin | 17 | 0.60 (0.20, 1.8) | 17 | 0.60 (0.20, 1.9) | 14 | 0.50 (0.20, 2.3) |
| ALT | 17 | 72 (12, 275) | 17 | 35 (11, 327) | 14 | 59 (8.0, 167) |
| GGT | 17 | 215 (21, 720) | 13 | 60 (10, 328) | 9 | 57 (17, 312) |
| ESR/WSR | 16 | 24 (1.0, 109) | 9 | 15 (2.0, 66) | 9 | 15 (2.0, 140) |
| **ALP** | **17** | **383 (124, 980)** | **17** | **180 (64, 1079)** | **14** | **206.5 (69, 918)** |

Total Bilirubin, ALT—alanine aminotransferase, GGT—gamma glutamyl transferase and ESR/WSR- erythrocyte sedimentation rate, ALP—alkaline phosphatase; median and range of lab parameters at baseline, six months and one year after OV therapy.

The median dose was 1000 mg, with the 500–1500 mg dose range identified among 47% patients with normalization of GGT within a year of therapy. There was no added benefit noted in patient 11 whose OV dose was titrated up to 3000 mg for 7.5 years. Patient 11 ultimately underwent orthotopic liver transplant due to severe portal hypertension and cirrhosis. The post-transplant period was complicated by recurrence of PSC as well as an

inflammatory pseudotumor of the liver for which 1500 mg OV was restarted. There were no reported adverse effects from OV throughout the entire course of therapy.

Except for Patient 16, all patients included in this study were on multiple PSC and IBD-specific therapies throughout the course of their disease process. Twelve patients received ursodiol therapy daily as first-line before initiation of OV and were continued on both therapies even after the initiation of therapy with OV. Only patients 3, 5, 10, 12, and 17 were treated with immunomodulators/biologic therapies (infliximab and adalimumab) for their concomitant diagnosis of IBD. Patients 1, 3, and 17 received methotrexate and 11 patients (65% of the cohort) were on ASA/mesalamine therapy, while patients 1 and 8 were on azathioprine for a brief period.

Patients 3, 8, 10, 12 and 16 were lost to follow-up after a one-year period. Patients 4 and 10 were enrolled in a clinical trial for OV in a different institution for a period of six months and three years, respectively, after which OV was discontinued. They continued to be monitored clinically as well as with labs and imaging at our institution. Patient 4 had evidence of severely dysplastic early hepatocellular carcinoma and atrophic hepatic cirrhosis on imaging which was initially managed with chemoembolization. This patient completed a six-month clinical trial of OV with normal GGT levels. However, given the rapid progression of his liver disease, the patient ultimately underwent liver transplantation three years after completion of OV therapy. Patients 3, 11 and 14 underwent liver transplantation at three, 7.5 and two years, respectively, after the initiation of OV. Patient 17 was non-compliant with the prescribed OV regimen.

### 3.4. Post-OV Therapy Characteristics

3.4.1. Post-OV Clinical Features

Clinical features were re-assessed following OV therapy with improvement noted in all patient-reported symptoms. At one-year post-OV therapy, clinical symptoms of fatigue declined in 93.8% cases with an improvement in jaundice, pruritus and anorexia noted in 81.3% patients (see Figure 1).

3.4.2. Post-OV Biochemical Laboratory Trends

Post-OV, biochemical laboratory values for TB, ALT, GGT, ESR/WSR and ALP were closely followed (See Table 3) at six and 12 months after the initiation of OV. Change in median laboratory values for TB, ALT, GGT and ESR/WSR from baseline to six months and then one year were assessed using Wilcoxon signed rank tests (see Table 4). The laboratory trends were then continuously followed on a yearly basis from one-year post-OV therapy, as available.

**Table 4.** Change in laboratory values for all included patients from baseline to 6 and 12 months.

| Changes From Baseline to 6 Months and 1 Year Post OV | | | |
|---|---|---|---|
| **Label** | **N** | **Median (Min, Max)** | ***p* Value \*** |
| Change in ALT, baseline to 6 months post-OV | 17 | −70.6 (−274.4, −11.7) | <0.001 |
| Change in GGT, baseline to 6 months post OV | 17 | −144.0 (−688.0, 52.0) | <0.001 |
| Change in ESR/WSR, baseline to 6 months post OV | 12 | 46.5 (−12.0, 309.0) | 0.003 |
| Change in Total Bilirubin, baseline to 6 months post OV | 9 | 14.4 (1.2, 64.3) | 0.004 |
| Change in ALT, baseline to 1 year post OV | 14 | −68.0 (−251.6, −11.8) | <0.001 |
| Change in GGT, baseline to 1 year post OV | 14 | −138.5 (−482.0, 5.0) | <0.001 |
| Change in ESR/WSR, baseline to 1 year post OV | 9 | 38.0 (−3.0, 270.0) | 0.016 |
| Change in Total Bilirubin, baseline to 1 year post OV | 9 | 14.8 (1.4, 139.0) | 0.004 |

\* Wilcoxon signed rank test for change.

Within the first year of OV therapy, TB levels improved in 52.9% of the patients (n = 9/17), median change of 14.8 (*p* = 0.004). The median TB value declined from 0.6 (n = 17) at baseline to 0.5 (n = 14) at one year after therapy (see Tables 3 and 4 and Figure 2).

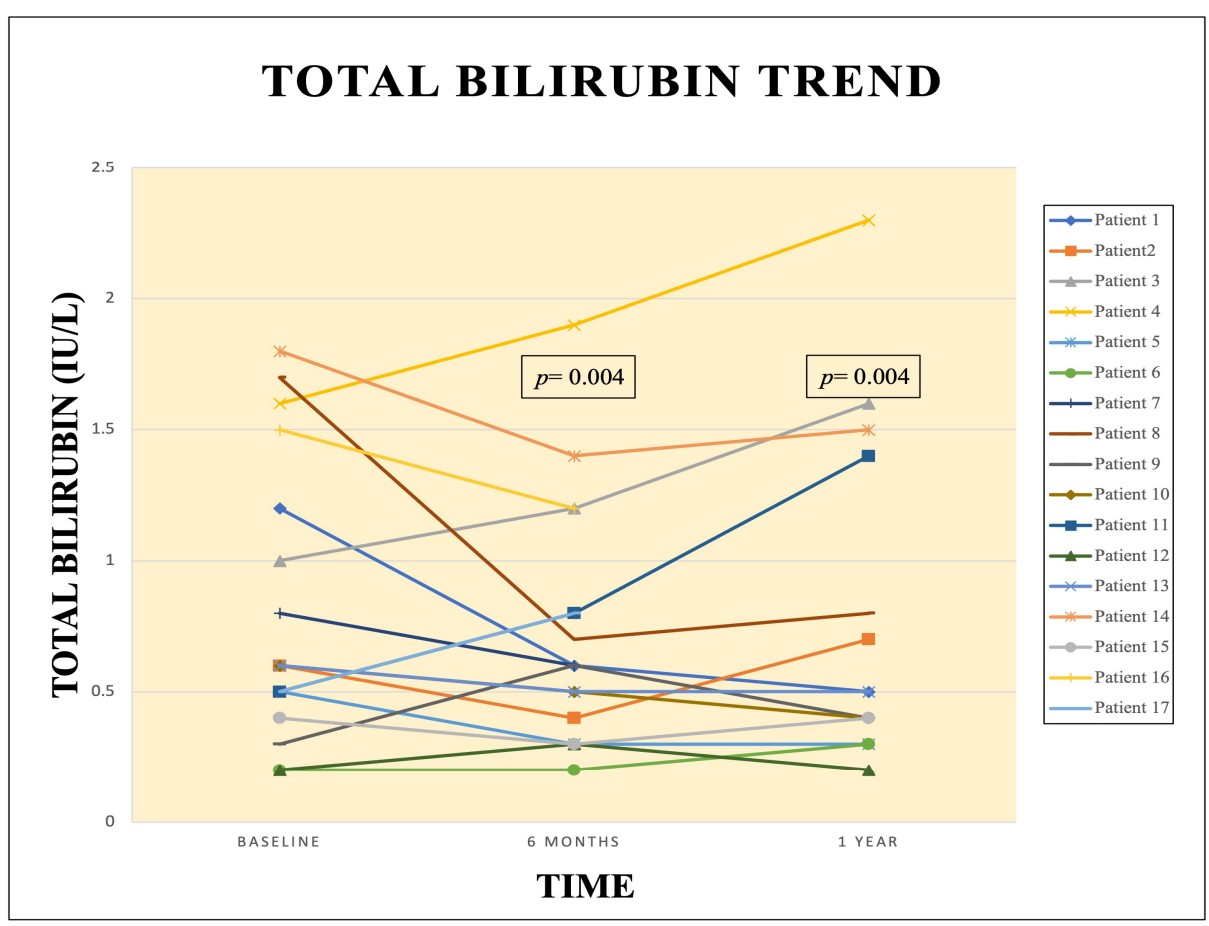

**Figure 2.** Total Bilirubin trends from baseline to 6-months and 1 year after initiation of Oral Vancomycin Therapy (OVT), *p* = 0.004.

GGT levels decreased in 76.9% of the cohort (n = 10/13) and normalized in 53.8% (n = 7/13) at six months of OV with a median of 60 U/L. At one year post-OV, GGT decreased as well as normalized among 55.6% (five out of nine patients) with a median of 57 U/L (see Tables 2 and 3). Median change in GGT from baseline to six months and one year was −144 (*p* < 0.001) and −138.5 (*p* < 0.001), respectively (see Table 4 and Figure 3). No follow-up GGT levels were available for four patients after the initiation of OV therapy. Among the 13 patients for whom GGT levels were followed, 61.5% (n = 8/13) returned to the normal range during the therapeutic course.

Median ALP levels declined to 180 U/L at six months (n = 17) and 206.5 U/L at one-year post-OV therapy (see Table 3). ALP levels declined from 29.4% of the cohort (n = 17) at baseline to 21.4% (n = 3/14) at one year after OV. AST levels decreased to normal among 47% (n = 8/17) patients at six months and continued to remain normal in 50% (n = 7/14) at one year of therapy.

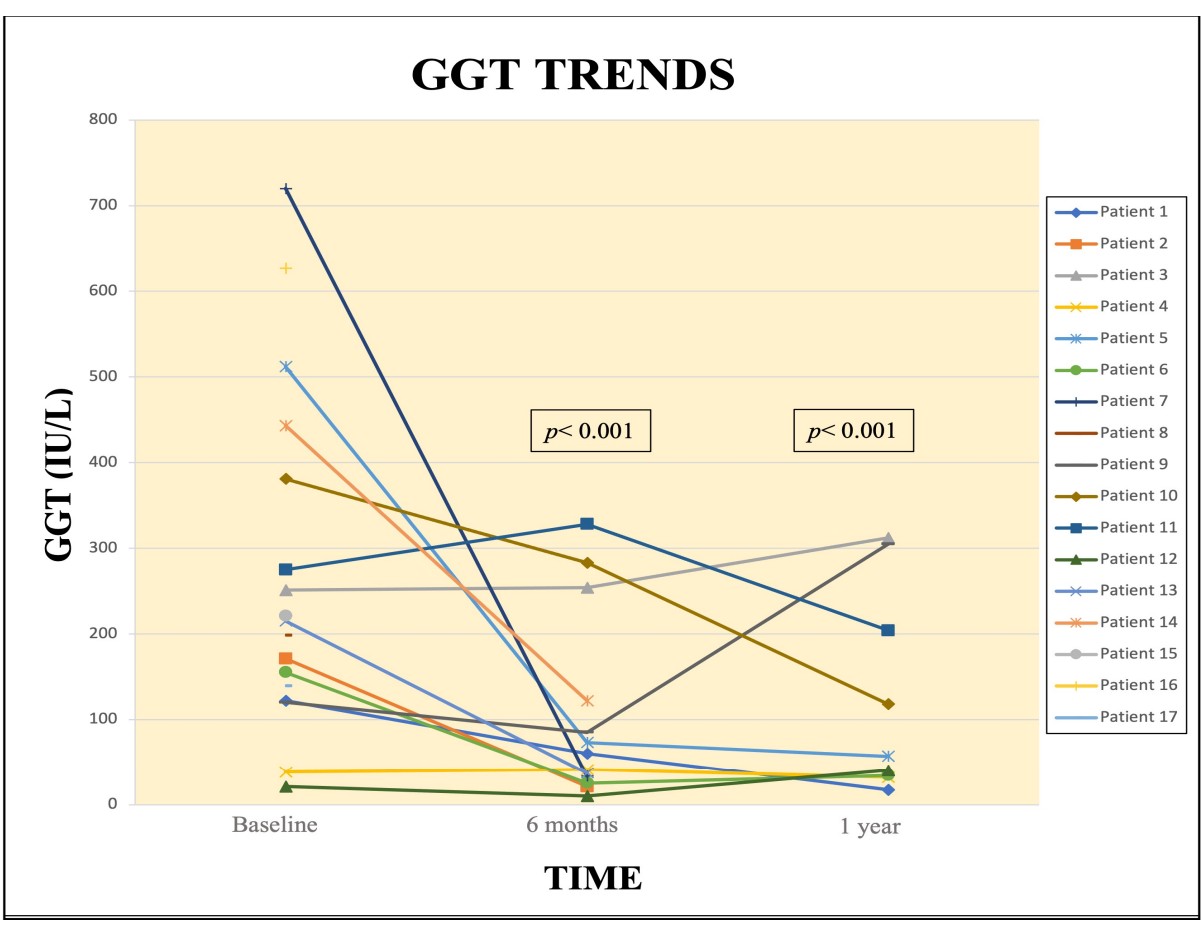

**Figure 3.** Gamma Glutamyl Transferase (GGT) Trends at baseline, 6 months and 1 year after initiation of Oral Vancomycin Therapy (OVT), *p* < 0.001.

ALT levels normalized in 58.8% patients (n = 10/17) and then 42.8% (n = 6/14) at six months and one year of OV, respectively. There was a decrease in ALT level amongst 76.4% (n = 13/17) at six months and 78.5% (n = 11/14) at one-year post OV. Median change in ALT from baseline to six months and one year was −70.6 (*p* < 0.001) and −68 (*p* < 0.001), respectively (see Table 4). ALT levels continued to decrease and ultimately normalized in 11 patients, i.e., 64.7% from baseline, through the entire therapeutic course beyond one year (see Tables 2 and 3 and Figure 4).

3.4.3. Post-OV Imaging Findings

All 17 patients had evidence of PSC on MRCP prior to initiation of OV. At six months post-OV, MRCP revealed improved PSC in 14.3% (n = 1/7), stable PSC in 57.1% (n = 4/7) and no evidence of PSC in 14.3% (n = 1/7). This was followed by stable PSC findings on MRCP among 50% (n = 6/12) with improvement noted in 8.3% (n = 1/12) and no evidence of PSC in 8.3% (n = 1/12) at one year post-OV. New PSC findings on MRCP at one year were noted among 33.3% (n = 4/12). Stable or absent PSC findings were evident on US for 85.7% (n = 6/7) patients with follow-up imaging data at both six months and one year after OV. No new PSC findings were noted on US one year after OV within this cohort and there was improvement noted for one patient. New findings of stricture with progressive dilation of the intrahepatic biliary duct on MRCP were noted about one year after discontinuation of OV in Patient 3. No follow-up imaging data was available for four patients (Patients 7, 8, 12, and 17) after OV therapy.

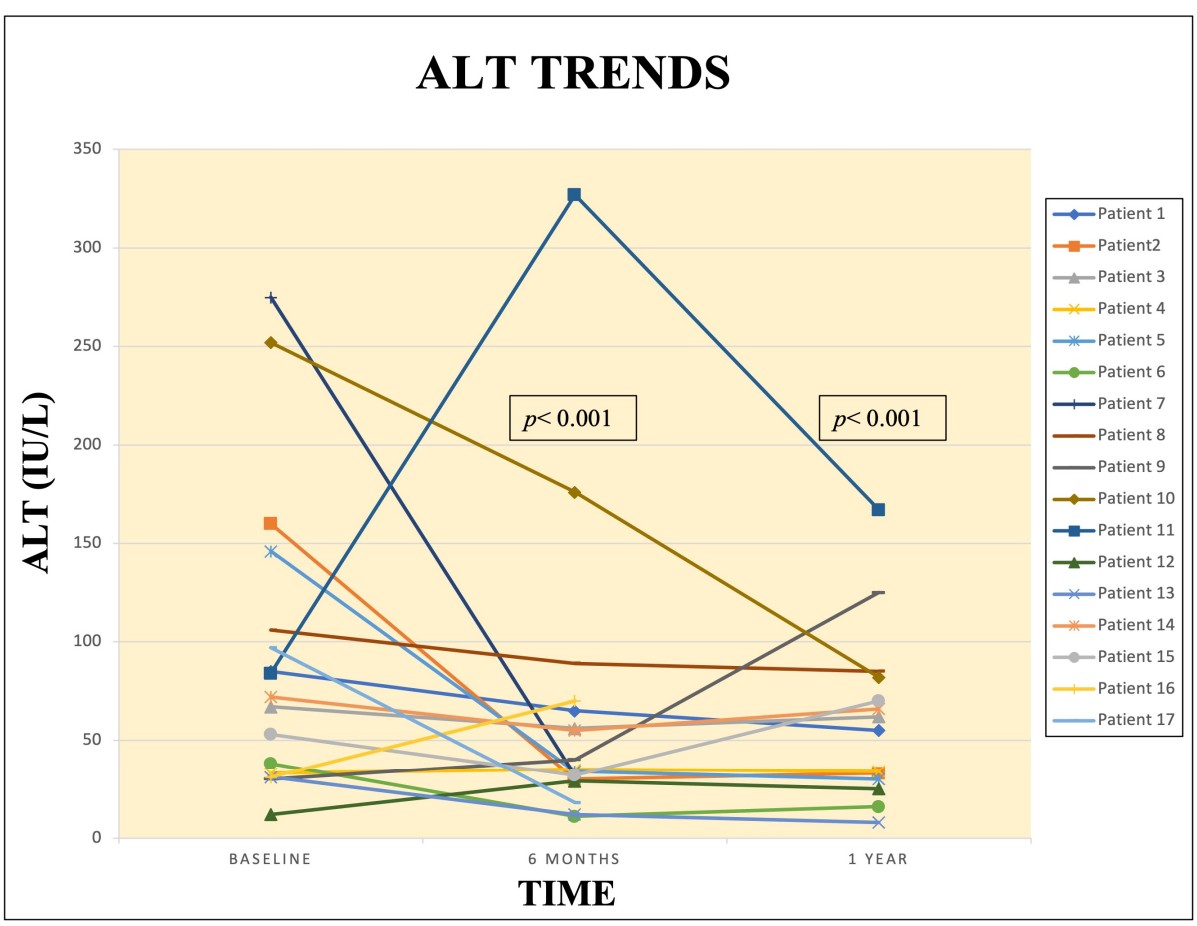

**Figure 4.** Alanine Transaminase (ALT) Trends at baseline, six months and one year after the initiation of Oral Vancomycin Therapy (OVT), *p* < 0.001.

## 4. Discussion

This descriptive case series highlights improvement in GGT and ALT over the course of therapy along with a 66.7% improvement of PSC findings (eight out of 12 patients) based on MRCP imaging within the first year of therapy, without any known adverse effects. The results highlight early change in biochemical as well as imaging findings in this slow, yet progressive disorder. We postulate that the improvements noticed were a result of the immunomodulating local effects of OV within the gastrointestinal and hepatobiliary tract. Four patients progressed to require liver transplantation irrespective of dosage or duration of OV. There was no significant alteration of the disease course in patients based on dosing, duration of OV or concomitant immunomodulatory therapy.

Oral vancomycin is a bactericidal antibiotic effective against gram-positive bacteria. It concentrates within the intestine with minimal oral absorption exerting dual action as an antimicrobial and immunomodulator by reducing cytokine release from T cells [5,16]. The rationale behind OV use is based on the dysbiosis within the normal intestinal microbiome identified in PSC patients and its ability to act as an immunomodulating antibiotic noted in small cohorts [17]. Davies et al. and Abarbanel et al. explored the positive effects of OV (initial dose 50 mg/kg/day) in two separate small uncontrolled case series of 14 patients each with promising results marked by normalization of GGT levels and symptom resolution [17,18]. Several studies indicate that clinical symptoms resurfaced and GGT levels increased upon withdrawal of OV with subsequent improvement on resuming OV therapy [18–20]. There is evidence of an improvement in overall prognosis for patients with GGT reductions below 50 U/L [19] with or without >25% decrease in GGT within the first year of PSC diagnosis [2,10,21].

There have been two adult studies showing significant decrease in ALP and clinical symptoms in PSC, describing symptom resolution as well as normalization of labs after administration of OV [22]. A triple blinded randomized control trial in the adult population by Rahimpour et al. revealed acceptable efficacy of OV in treatment of PSC among adults based on a significant decline in the mean PSC Mayo risk score as well as the ALP level at three months post-therapy [23]. As evidenced in large retrospective reviews of pediatric PSC, ALP is only elevated in about 53–81% patients compared to GGT elevations among 94–100% patients at the time of PSC diagnosis, making this marker unreliable among children [1,7]. Ali et al. report a 49% and 62.2% normalization of GGT and ALT respectively, among 45 pediatric PSC patients on OV within six months of therapy. The cohort comprised adult and pediatric patients (n = 59) and revealed a significant improvement in hepatic biochemistry during the median 2.5 years of OV therapy [20]. Cox et al identified a significant improvement in GGT, ALT, and ESR with resolution of inflammation and fibrosis on liver histology in a cohort of 14 pediatric patients treated with OV, likely related to immunoregulatory effects with increased serum TGF-b and circulatory regulatory T cell levels, identified in a subsequent study by the same group [17,22].

The Pediatric PSC Consortium recently conducted the largest retrospective pediatric PSC study with 264 matched patients (88 patients in each treatment/placebo cohort) and identified GGT normalization at one year of therapy with OV, ursodiol and placebo without any statistically significant difference between the three groups. This study identified an increased spontaneous rate of biochemical normalization among patients with milder disease phenotype, early stage of PSC and/or fibrosis in the absence of any known PSC therapies. This supports the theory that GGT fluctuations are common during the early stages of PSC, thus highlighting the lack of effectiveness of the two most utilized therapies for this disorder [24]. The exact therapeutic effect of OV remains elusive specifically with respect to differences in disease progression based on its efficacy at different stages of fibrosis affecting the hepatobiliary tract and its role as primary versus salvage therapy in advanced PSC not responding to other known treatment modalities [19].

No adverse events including vancomycin resistant enterococcus (VRE) were reported in our study, the safety profile of OV being similar to related case series evident from literature review. There is a dearth of data in this aspect, given the absence of standardized surveillance testing for VRE in the population irrespective of the duration of OV therapy. A prospective adult study revealed development of VRE in over 20% patients with 10 days of OV therapy for Clostridium difficile infection [15], making this a concern in the long term for PSC patients given their chronic disease process and need for multiple medication regimens, specifically surrounding the liver transplant period, based on the severity of disease.

*Limitation*

The major limitations of this study are its retrospective nature, small size of the cohort, lack of long-term follow-up and patients being lost to follow-up, the absence of a control group, and OV dose variation between patients. The slow progressive nature of the disease as well as associated comorbidities and absence of standardized biomarkers makes identification of medical therapies challenging. There are several unaccounted confounders related to the use of concomitant targeted therapies for IBD and degree of disease states. The clinical course and response of IBD with respect to PSC therapies was not described in this case series. Close monitoring of adherence to prescribed therapies was difficult and there was no standardized routine surveillance conducted to assess for VRE infections.

**5. Conclusions**

In conclusion, we identified improvement of PSC based on biochemical laboratory and imaging findings while on OV therapy within this cohort. However, there is little evidence to distinguish between the natural course of PSC progression and the therapeutic impact of OV. The elusive nature of this rare disease process, along with the utilization of multiple overlapping therapies including antibiotics and immunomodulators for treatment

of IBD as well as PSC, remain barriers to the identification and development of safe and effective therapeutic strategies for PSC. Furthermore, future research, including randomized controlled trials, is warranted to identify effective and targeted therapeutic strategies for this progressive disorder specific to the pediatric population.

**Author Contributions:** Conceptualization, A.J.A., S.P., L.M., J.A.K., V.H., K.R. and M.N.K.; Methodology, A.J.A., S.P., L.M., J.A.K., V.H., K.R. and M.N.K.; Validation of Data, A.J.A., S.P., L.M., J.A.K., S.W., V.H., K.R. and M.N.K.; Formal Analysis, A.J.A., S.P., L.M., J.A.K., S.W., V.H., K.R. and M.N.K.; Data Curation, A.J.A., S.P., L.M., J.A.K., S.W., V.H., K.R. and M.N.K.; Writing—Original Draft Preparation, A.J.A. and M.N.K.; Writing—Review and Editing, A.J.A., S.P., L.M., J.A.K., S.W., V.H., K.R. and M.N.K.; Supervision, S.P., L.M., J.A.K., V.H., K.R. and M.N.K.; Visualization; A.J.A., S.P., L.M., J.A.K., V.H., K.R. and M.N.K.; Software Utilization, A.J.A., S.W. and M.N.K. All authors have read and agreed to the published version of the manuscript.

**Funding:** This research received no external funding.

**Institutional Review Board Statement:** The Institutional Review Board (IRB) performed an expedited review of the manuscript proposal and deemed it as exempt research, i.e., minimal risk research using/involving secondary research which does not require consent, and waived the need for continuing review.

**Informed Consent Statement:** Patient consent was waived due to the study deemed as "minimal risk research using/involving secondary research which does not require consent" per the Institutional Review Board.

**Data Availability Statement:** Data available on request due to restrictions. The data presented in this study are available on request from the corresponding author. The data are not publicly available due to privacy and ethical restrictions.

**Acknowledgments:** Department of Pediatric Gastroenterology, Hepatology and Nutrition at Cleveland Clinic Children's.

**Conflicts of Interest:** The authors declare no conflict of interest.

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
