# Peer review of "Single Center Experience of Oral Vancomycin Therapy in Young Patients with Primary Sclerosing Cholangitis: A Case Series"

_livers, doi:10.3390/livers3010009_

Round 1
Reviewer 1 Report
The manuscript entitled " SINGLE CENTER EXPERIENCE OF ORAL VANCOMYCIN THERAPY IN YOUNG PATIENTS WITH PRIMARY SCLEROSING CHOLANGTIS: A CASE SERIES" presented by Alenchery et al, summaries a case study on effectivity of Vancomycin over primary sclerosing cholangtis. Overall, the manuscript is addressing and delivering the scientific content. The manuscript is written well and considered all scientific protocol. However, i did not find ethical permission statement, which need to be mentioned as per requiremen.
Some minor exercise is required to improve further its content.
· Please add current application of Vancomycin in Introduction.
· Any similar study that include similar chemical scaffold or structure must be discussed.
· Any other future studies that need to warranted must be concluded for conversion of this value addition or repurposing to certain market product.
· Language and any other typological mistake can be address
· Please check pattern of reference as per format.
Reviewer 2 Report
I have read this case series with great interest and I believe that it will also be of great interest to the readers of this journal.
In my opinion, the presented results are the basis for a randomized clinical trial evaluating the effects of OV in PSC.
I would suggest, however, slightly changes the layout of the work, as some of the results were included in the material and methods. However, the description of the applied intervention should be moved to the methods.
Nevertheless, the observation was carried out in growing children, I would like the authors to present the effect of OV on the concentration of ALP.
Reviewer 3 Report
The authors insisted the improvement of PSC based on biochemical laboratory and imaging findings while on OV therapy within this cohort. However, there is little evidence to distinguish between the natural course of PSC progression and the therapeutic impact of OV.
I have some comments.
1.The mechanism by which vancomycin treats PSC is unknown. It is also unclear how long the treatment should last.
2. It is also necessary to describe how UC and CD changed with vancomycin administration.
3. As the authors also describe, it is difficult to explain the difference between the effects of natural course and OV. It seems that there is not enough evidence to conduct an RCT.
Round 2
Reviewer 3 Report
The mechanism of therapeutic effect of OV on PSC seems to be largely unknown. Through the text and comments of this revised version, the authors have sincerely answered my point. I consider this revised version acceptable.